# Lipid Remodeling Confers Osmotic Stress Tolerance to Embryogenic Cells during Cryopreservation

**DOI:** 10.3390/ijms22042174

**Published:** 2021-02-22

**Authors:** Liang Lin, Junchao Ma, Qin Ai, Hugh W. Pritchard, Weiqi Li, Hongying Chen

**Affiliations:** 1Germplasm Bank of Wild Species, Kunming Institute of Botany, Chinese Academy of Science, Kunming 650201, China; linliang@mail.kib.ac.cn (L.L.); majunchao@mail.kib.ac.cn (J.M.); aiqin@mail.kib.ac.cn (Q.A.); h.pritchard@kew.org (H.W.P.); 2Royal Botanic Gardens, Kew, Wellcome Trust Millennium Building, Wakehurst Place, West Sussex, Ardingly RH17 6TN, UK

**Keywords:** cryopreservation, cryoprotectant agents, ex situ conservation, lipid, Magnoliaceae, plant vitrification solutions

## Abstract

Plant species conservation through cryopreservation using plant vitrification solutions (PVS) is based in empiricism and the mechanisms that confer cell integrity are not well understood. Using ESI-MS/MS analysis and quantification, we generated 12 comparative lipidomics datasets for membranes of embryogenic cells (ECs) of *Magnolia officinalis* during cryogenic treatments. Each step of the complex PVS-based cryoprotocol had a profoundly different impact on membrane lipid composition. Loading treatment (osmoprotection) remodeled the cell membrane by lipid turnover, between increased phosphatidic acid (PA) and phosphatidylglycerol (PG) and decreased phosphatidylcholine (PC) and phosphatidylethanolamine (PE). The PA increase likely serves as an intermediate for adjustments in lipid metabolism to desiccation stress. Following PVS treatment, lipid levels increased, including PC and PE, and this effectively counteracted the potential for massive loss of lipid species when cryopreservation was implemented in the absence of cryoprotection. The present detailed cryobiotechnology findings suggest that the remodeling of membrane lipids and attenuation of lipid degradation are critical for the successful use of PVS. As lipid metabolism and composition varies with species, these new insights provide a framework for technology development for the preservation of other species at increasing risk of extinction.

## 1. Introduction

Cryopreservation at ultra-low temperature (e.g., liquid nitrogen, LN, −196 °C) is widely used for the long-term storage of viable cells, tissues, organs, or organisms of both animals and plants [1,2,3,4,5,6]. This technique has the potential to preserve the morphological, physiological, biochemical, and genetic properties of material as cellular metabolism is stopped [7]. Preventing intracellular ice formation and growth is crucial for cryopreservation success and this can be achieved most simply by reducing cell water content. Once intracellular water is reduced sufficiently, through osmotic- or freeze-dehydration, surviving cooling to very low temperature is less problematic [8]. However, imposed changes in cell water relations can be stressful, especially at the cell membrane level [3,9,10,11].

The earliest methods for plant cryopreservation employed controlled-rate-cooling; for example, for hardy plant tissues and cells that could withstand intracellular ice formation and freeze dehydration [12]. Pretreatment of cells with cryoprotectant agents (CPA), such as dimethyl sulfoxide (DMSO), reduces the level of dehydration damage [13]. The combination of dehydration, CPA, and cooling facilitates the formation of intracellular aqueous glasses, i.e., vitrification [2,10,12]. The efficiency of the cell vitrification process, and avoidance of ice formation, depends on the interplay between CPA concentration and the cooling and warming rates [14]. Such control is often more challenging for tropical plant material with a variety of cell types [15].

Vitrification is achieved by exposing cells and tissues to extremely concentrated plant vitrification solutions (7–8 M) of CPA, followed by solidification to a metastable glass via direct immersion in LN [16,17,18]. Although ice crystal formation is avoided, severe chemical dehydration can damage cells [16]; although this impact can be reduced by the pretreatment of tissues with high levels of sugar or sugar alcohol prior to exposure to PVS. The most commonly used plant vitrification solution is plant vitrification solution 2 (PVS2). The stepwise means of preservation is now well established (Figure 1): Osmoprotection (loading treatment; notation L), chemical dehydration with plant vitrification solution (e.g., PVS2; P); fast cooling (with LN; F); rapid thawing (T); dilution and rehydration (unloading treatment; U); and planting recovery in vitro (R). Potentially, each step results in the accumulation of stresses that can have severe consequences for cell survival.

The application of functional genomics and proteomics to cryopreservation research has been advocated [19,20,21]. Such approaches have revealed the genetic pathways leading to cold acclimation and freezing tolerance, and identified the involvement of key cold-regulated genes. Genes and proteins are also implicated in cryo success in the glycolytic and other metabolic pathways, particularly processes involved in dehydration tolerance, osmoprotection, and membrane transport [19]. The regulatory mechanism in *Arabidopsis* seedlings during the post-cryo response involves coping with reactive oxygen species (ROS) and oxidative stress [20]. During cumulative stress, the sites of action of ROS in plant and fungal cells are multifarious, with protection provided by scavenging enzymes and antioxidants such as glutathione [22]. In the membrane-associated proteins in eicosanoid and glutathione metabolism (MAPEG) superfamily of animal cells, glutathione transferase has both cytosolic and membrane sites, providing a structural basis for oxidative stress protection in membranes [23]. Moreover, plants responsive to cold acclimation accumulate cold-regulated (COR) proteins that can fold on drying and bind peripherally to membranes thorough interaction with galactolipids [21]. Membranes are key sites for injury during plant cell freezing [15]. Membrane shrinkage and separation from the rigid cell wall occurs during water loss-induced plasmolysis and lowering temperature induces phase changes that enable solute leakage [24]. Thus, an understanding of membrane architecture and composition and how they are affected during the cryopreservation process (i.e., osmoprotection, cryoprotection, cooling/warming) might provide insight to the mechanisms of tolerance and survival.

Plant membranes mainly consist of galactolipids (two classes), phospholipids (six classes), and lysophospholipids (three classes) [25]. Classes with large polar head groups, such as phosphatidylcholine (PC) and digalactosyldiacylglycerol (DGDG), tend to form bilayer lipids that may increase membrane stability, while classes with small head groups, such as phosphatidylethanolamine (PE), monogalactosyldiacylglycerol (MGDG), and phosphatidic acid (PA), predispose the membrane to form the hexagonal II phase [3,26,27]. Dehydration results in osmotic pressure change in the cell and this can lead to membrane remodeling. For example, increases in PA and unsaturated fatty acid levels are accompanied by decreases in MGDG during dehydration [28,29,30,31]. High PA is associated with dehydration-induced viability stress in desiccation-sensitive seeds, whilst inhibition of phospholipase Dα-mediated PA accumulation increases desiccation tolerance in recalcitrant seeds [27]. Consequently, we hypothesized that membrane lipid remodeling could exacerbate or alleviate some of dehydration and low temperature stresses during cryopreservation. There is some evidence for this in the green algae *Chlamydomonas reinhardtii*; drastic degradation of lipids occurs after two-step freezing to liquid nitrogen temperature, but lipid remodeling occurs following thawing [3]. Similarly, the composition of membrane phospholipids changes in Western Australian plant species during alternating temperature pre-conditioning, and this is thought to contribute to successful cryopreservation [32]. However, a detailed dissection of lipid remodeling during all steps of plant cell cryopreservation and recovery appears to be lacking.

To explore this issue, we took a lipidomics approach using electrospray ionization mass spectrometry (ESI–MS/MS) lipid profiling. This enabled the quantification of changes in the molecular species of membrane lipids during cryopreservation. We also established embryogenic cells (ECs) of *Magnolia officinalis* as the model system for cryopreservation. Using ECs rather than shoot tips or meristems as an experimental system has many advantages. The physiological state of ECs can be made relatively uniform through control of the growth conditions. In addition, ECs viability can be detected easily using confocal microscopy. Finally, ECs are invaluable to the preservation of species with non-bankable seeds. There are seed storage assessments for only a handful of >300 species in the genus *Magnolia* (328 accepted names; http://www.plantsoftheworldonline.org, accessed on 18 January 2021), and seed tolerance of drying varies from more (eight species) to less (three species) likely (https://data.kew.org). As it is estimated that nearly 40% of plant species are at risk of extinction (http://www.weforum.org), the design of cryobiotechnologies for conservation based on fundamental principles is urgent [33]. In the present study, we found that each step of a PVS cryoprocedure affects lipid remodeling in ECs in specific ways that enable the retention of cell integrity and regrowth capability.

## 2. Results

### 2.1. Stepwise Cryopreservation of Embryogenic Cells (ECs)

A PVS2-based vitrification cryopreservation protocol for plant tissue, as indicated in Figure 1, was applied to ECs of *M. officinalis.* It included up to seven steps. As it was not possible to separate thawing from the liquid nitrogen cooling step, samples were taken at six steps (Treatments 1–6) and the ECs were immediately treated with hot (75 °C) isopropanol to stop any lipid metabolism. To separate out the effects of individual steps in the overall cryopreservation process, single steps were excluded from the protocol (Treatments 7–12) and the ECs were treated in the same way.

During cryopreservation, and after PVS2 (CLP) and freezing treatments (CLPFT), the green fluorescence intensity signifying live cells decreased significantly by about 60% (Figure 2). But after the recovery step (CLPFTUR), the green fluorescence level gradually increased and the surviving ECs grew. The samples for all missing cryopreservation steps were taken after 2 weeks in the recovery treatment. However, excluding the PVS2 treatment (CLPFTUR/-P), and even using loading buffer to compensate for PVS2 treatment time (CLPFTUR/-P+L), resulted in lethal damage such that no tissue survived (Figure 2). These results showed that treatment with PVS2 is much more critical for successful cryopreservation than the loading step.

### 2.2. Profiling of the Molecular Species of ECs’ Membrane Lipids during Cryopreservation Treatments

To explore the changes in membrane lipids and how each cryopreservation step affected them, we harvested ECs of *M. officinalis* for lipidomics analysis using ESI-MS/MS. It allowed us to identify >120 diverse polar glycerolipids, including major phospholipid classes (PC, PE, phosphatidylinositol (PI), and phosphatidylglycerol (PG)), galactolipid classes (MGDG and DGDG), and minor classes (PA, phosphatidylserine (PS), lysophosphatidylcholine (Lyso PC), lysophosphatidylethanolamine (Lyso PE) and lysophosphatidylglycerol (Lyso PG)). Each molecular species was identified in terms of the total number of acyl carbon atoms and double bonds [34].

Lipid levels and composition of ECs differed with the cryopreservation step (Figure 3 and Appendix A and Table 1 and Appendix A). For example, the levels of most lipids increased after PVS2 treatment (CLP) but decreased after freezing–thawing (CLPFT). However, neither the patterns of lipid composition, the PC/PE ratios, nor the acyl chain length (ACL) or double bond index (DBI) changed significantly (Figure 4, Appendix A). This indicates that the ECs membranes maintained good integrity and fluidity throughout. However, total lipids degraded dramatically when PVS2 treatment was excluded, i.e., CLPFTUR/-P and CLPFTUR/-P+L (Table 1, Figure 3). The lipid composition pattern of those two treatments also showed dramatic differences from other treatments (Figure 4 and Appendix A). Principal component analysis (PCA) of absolute and relative levels of glycerolipids also supported that lipid component differs between treatments including and excluding PVS2 (Appendix A). Moreover, changes in PC and PE contributed to the resilience of ECs to cryopreservation and showed that each step in the process impacts differently on membrane lipid composition and thus on membrane integrity (Appendix A, Appendix A).

### 2.3. Loading Treatment Induced Membrane Lipid Turnover

The loading treatment (2 glycerol and 0.4 M sucrose) is considered as an osmoprotection step, which minimizes the osmotic damage caused by the later addition of full strength PVS2 [16]. However, how this treatment affects membrane integrity is unclear. Possibly due to the short treatment time (20 min), loading (CL) changed neither the total lipids (Table 1), nor acyl chain length (Appendix A) nor double-bond index (Appendix A). Interestingly, the PA level increased (66%) significantly from 6.04 to 10.04 nmol/mg, and PG by 54% from 2.28 to 3.52 nmol/mg (Figure 5, Table 1). The composition of PC and PE decreased slightly, while PA significantly increased from 10.3% to 17.8%, and PG increased from 3.9% to 6.1% (Appendix A). Surprisingly, increases in PA 34:2, PA36:4, PG 34:2, and PG 36:4 corresponded stoichiometrically to decreases in PC 34:2, PC 36:4, PE 34:2, and PE 36:4 (Figure 6a). These changes indicated that some turnover of lipid molecular species from PC and PE to PA and PG may have occurred by the end of the loading treatment. Not only would a decreased lipid fluidity result, but the increase of PA might transduce desiccation stress [35]. Although lysophospholipids only occupied approximately 1% of total lipids under normal growth conditions, they increased by 72.3% after loading (Table 1, Appendix A). Lyso PE significantly increased (0.13 to 0.29 nmol/mg), similar to the rapid and considerable changes reported when plants are exposed to a range of biotic and abiotic stimuli [36].

To analyze the effects of loading treatment on membrane lipids, we compared CLPFTUR treatment (standard protocols) with CLPFTUR/-L treatment, which lacked the loading step (Figure 1 and Table 1). Total lipids significantly decreased by 72.6% in CLPFTUR/-L treatment, especially PC and PE (Table 1). Although PC was still the main lipid component (Figure 4 and Appendix A), the survival of ECs was significantly lower compared with standard protocols (Figure 2). We then analyzed whether loading only plays a role in dehydration by extending PVS2 treatment to 35 min (CLPFTUR/-L+P). Total lipids levels increased from 20.1 to 43.5 nmol/mg and the composition of lipids returned towards normality, as defined by the standard protocol (CLPFTUR) (Figure 4, Table 1 and Appendix A).

These results suggested that the loading treatment had two effects, immediate adjustment to mild dehydration and downstream osmoprotection. Loading protects the membrane by inducing membrane lipid turnover, with the increase of PA potentially signaling desiccation stress that, in turn, prepares the membrane for more extreme chemical dehydration by PVS2. Without the loading step, total lipids significantly decreased, and EC survival was low. Although longer exposure time to PVS2 increased total membrane lipid somewhat, and EC survival significantly increased (Figure 2), the function of loading to membrane integrity during cryopreservation was irreplaceable.

### 2.4. PVS2 Treatment Induces Lipid Membrane Remodeling and Results in Total Lipid Increase

Full strength PVS2 (7.98 M) was used as a chemically-induced dehydration step to reduce the risk of lethal ice formation during cooling/warming to/from liquid nitrogen temperature. Compared with the loading treatment, the application of PVS2 for 15 min (CLP) significantly increased total lipids by 32%, from 57.3 to 75.6 nmol/mg (Table 1). Changes in phospholipids, such as PI (5.7 to 8.7 nmol/mg), PE (8.5 to 12.2 nmol/mg), and PC (23.6 to 34.4 nmol/mg), contributed 95% of this increase. Notably, the PC content after PVS2 treatment was 6.4 nmol/mg higher than in the control (C). The increases in PC and PE did not result from decreases in PA and PG, which showed no change during this step.

To examine the PVS2 effect on the membrane, we also compared CLPFTUR treatment (standard protocol) with CLPFTUR/-P treatment (i.e., no PVS2 step; Figure 1, Table 1). Without PVS2 dehydration, total lipids decreased by 92.8%, from 73.4 to 5.3 nmol/mg (Figure 3 and Figure 7b, Table 1). Moreover, the levels of eight classes of lipids, especially PC and PE, showed significant decreases (Table 1). The dramatic lipid decrease may relate to intracellular ice formation physically disrupting the membrane. We then wondered whether extending the loading treatment could partially replace the PVS dehydration step. When loading for 35 min (CLPFTUR/-P+L) was compared with CLPFTUR/-P, no increase in total lipid was observed (Table 1). All the lipids showed the same decrease as CLPFTUR/-P treatment (Figure 6) and the patterns of lipid composition varied from the standard cryopreservation process (CLPFTUR) (Figure 4).

These results suggested that PVS2 induces lipid membrane remodeling by increasing total lipids, especially PC and PE, which might improve membrane fluidity. The increase of PC and PE did not result from lipid turnover. In addition, this treatment also serves as an important dehydration step in the cryopreservation protocol, without which lethal intracellular freezing and membrane deterioration would likely occur during cooling/warming.

### 2.5. The Effect of Cooling and Warming on Membrane Lipids

After adjusting the membrane composition, ECs in PVS2 solution were immersed in liquid nitrogen overnight, and then thawed in a water bath at 40 ± 1 °C for 2 min. Total lipids of ECs decreased from 75.6 (CLP) to 68.3 nmol/mg (CLPFT), mainly as a result of reductions in PC (34.4 to 27.7 nmol/mg), PE (12.2 to 9.0 nmol/mg), and PI (8.7 to 6.7 nmol/mg). Decreases in these lipids were partially converted into increases in PA (from 10.6 to 13.3 nmol/mg), PG (3.6 to 5.2 nmol/mg), DGDG (2.1 to 2.8 nmol/mg), and LysoPG (0.23 to 0.29 nmol/mg). These changes represented a second increase after freezing and thawing (cold stress) beyond that determined after the loading treatment (CL) alone (Figure 6, Table 1). Indeed, there was a general similarity in remodeling between PE, PC, and PA, PG after loading (CL), and the freezing/thawing (CLPFT) steps.

Freezing/thawing alone (CLPFTUR compared with CLPFTUR/-FT) did not change the total lipid level, at 73.4 and 73.9 nmol/mg, respectively (Table 1). However, the lipid composition in these two treatments were different (Figure 4, Appendix A), with PA and PG up to 9% higher and PC, PE, and PI up to 5% lower in the CLPFTUR/-FT treatment (Appendix A). Interestingly, the results showed that PA levels in the control (C) increased to 10.0 nmol/mg after the loading treatment (CL), and fell back to control levels after the entire CLPFTUR treatment (4.2 nmol/mg). However, when the cryopreservation process lacked the LN freezing/thawing step (CLPFTUR/-FT), the PA accumulation remained high (10.8 nmol/mg) (Table 1). These findings indicated that the processes of cooling and warming causes the remodeling of particular membrane lipids.

### 2.6. Lipid Membrane Composition Readjusted after Unloading

CLPFTU treatment (unloading for 20 min) used 1.2 M sucrose to replace 7.98 M PVS2 after the freeze/thaw step. Total lipids increased from 68.3 (CLPFT) to 73.3 nmol/mg (CLPFTU) (Table 1), and were accompanied by combined increases for PC, PE, and PI of 9.7 nmol/mg, and combined decreases of 4.5 nmol/mg for PA and PG. Interestingly, the decrease in PA 34:2, PA36:4, PG 34:2, and PG 36:4 corresponded stoichiometrically to PC 34:2, PC 36:4, PE 34:2, and PE36:4 increases, and these changes were opposite in nature to the loading treatment (CL) (Figure 6a). Thus, lipid membrane remodeling from loading and unloading had opposite trends (Table 1). When unloading was excluded (i.e., CLPFTUR/-U vs. CLPFTUR), total lipids (60.2 nmol/mg) and the levels of PS, PE, PC, and PA were lower, and PI was dramatic lower (Figure 5 and Figure 7, Table 1). In contrast, PG, MGDG, and DGDG were all slightly higher as a result of CLPFTUR/-U treatment (Table 1).

These results suggested that the unloading treatment facilitated the restabilization of the membrane by increasing total lipids and lipid turnover, potentially reducing the risk of the cells fully rehydrating too quickly if the unloading step was excluded.

## 3. Discussion

Whilst the successful use of vitrification-based cryopreservation has been reported for plant cells of a wide range of species over the last three decades, we still know little about the cellular processes and molecular adjustments that confer resilience to the multiple stressors imposed during the procedure. As for many other abiotic stresses, the cell membrane is a major site of injury during cryopreservation. But how each step of a PVS vitrification-based cryoprotocol contributes to maintain membrane integrity is little understood. To gain insight, we systematically compared the lipid molecular species profiles of *M. officinalis* ECs after each step in a vitrification-based cryoprotocol. Including the analyses when single steps in the protocol were omitted, we generated a total of 12 comparative lipidomics datasets.

We found that ECs have a membrane phospholipid (PL) composition typical of seeds [27], and substantially different to photosynthetic material, such as leaves and the green alga *C. reinhardtii*, which mainly contain galactolipids [3]. EC membranes have PC, PE, and PG as the main components, plus relatively small amounts of PS, PI, and PA (Appendix A). Similar lipid composition was observed in bilayer lipids (PC and DGDG), but PA contents (Appendix A and Appendix A) are higher than in seed lipid membranes [27]. In contrast, photosynthetic leaves mainly contain galactolipids rather than phospholipids (Appendix A). Moreover, the lipid membrane of the green alga *C. reinhardtii* contains 42% MGDG and almost no PC [3], compared to 4.5% MGDG and 48% PC in ECs (Appendix A). Such large differences in the initial lipid compositions of *M. officinalis* ECs and *C. reinhardtii* cells might suggest the need to tailor the design of cryopreservation protocols to cope with the likely variance in membrane thermal and hydraulic conductivity properties of the two biological systems. Perhaps the best examples of this is the preference of *C. reinhardtii* cells for methanol as a CPA during controlled rate cooling [3].

As membrane function under stress is mainly determined by lipid composition [25,27,37], some understanding of cellular membrane composition should benefit the optimization of a cryopreservation protocol. However, we observed that both the lipid content of ECs and the membrane composition was not static during cryopreservation, but remodeled at each step in the process to reflect the specific stress being imposed. Osmoprotection during the loading treatment stimulated substantial membrane remodeling by lipid turnover, with PA and PG increasing and PC and PE decreasing. The increase of PA might transduce desiccation stress to the cell membrane lipids, as observed in other stress responses [35]. Thereafter, PVS treatment induced a significant increase in total lipids, especially in PC and PE (Figure 7 and Table 1), and attenuation of lipid degradation. These molecular adjustments, including the reversal during unloading of those induced by loading, enabled the ECs to better survive freezing stress. Similarly, freezing tolerance in *Arabidopsis* plants need lipid remodeling at the outer chloroplast membrane [26]. As for plants cells in general under abiotic stress, we showed that two modes of response to cryo stress by ECs were mediated by lipids: As signaling molecules and for membrane remodeling.

Membrane remodeling is initiated during the loading step of PVS cryopreservation (Table 1), which is known to provide osmoprotection and increase dehydration resistance of tropical materials [16,32]. It is thought that the loading treatment enables the stabilization of membranes during moderate osmotic stress before severe dehydration in PVS2 [38], but how this treatment affects lipid turnover in such a short (20 min) time is unclear. We found that loading (CL) induced lipid turnover between PC 34:2, PC 36:4, PE 34:2, PE 36:4 and PG 34:2, PG 36:4, PA 34:2, PA 36:4. A consequence of these modifications was a significant increase in PA. Moreover, it is evident that the biochemical route for lipid turnover is precise and reversible, as the changes during loading were revoked during the unloading (CLPFTU) treatment (Figure 6, Table 1).

To maintain membrane functions under stress, plant cells can turn over head groups among the classes of lipid or adjust the degree of unsaturation. More double bonds increase lipid unsaturation, enhancing membrane fluidity, and maintaining membrane integrity. For example, double bond increases in PC are linked to greater survival in shoot tips of *Grevillea scapigera, Loxocarya cinerea,* and *A. thaliana* after PVS vitrification cryopreservation [32]. In this case, changes in DBI could be attributed to the 21-day period of pre-conditioning. Similarly, the longer times associated with acclimation, e.g., in sucrose, can result in DBI increase, which has a positive correlation with cryopreservation survival [39]. During PVS cryopreservation treatments though, any change in the degree of unsaturation could be limited by short treatment times and sources of energy. Loading and unloading treatment times of 20 min each were probably too short to allow alteration in membrane lipids DBI (Appendix A), because such metabolism involves complex reactions [40].

An alternative and energetically more efficient way of retaining membrane function is the turning over of head groups among the classes of lipid, as observed here in ECs (Figure 7 and Table 1). In plants, lipid turnover of glycerolipids head groups is an important part of lipid synthesis, degradation, and/or homeostasis [41]. Remodeled membrane lipids but maintenance of the degree of unsaturation are used to adapt to frequent temperature alterations [40]. After dehydration of *Arabidopsis thaliana* leaves, all degraded molecular species corresponded to the increases of PA molecular species with the same acyl structures; thus, the decrease of 34:6 MGDG specifically corresponded to the increase of 34:6 PA [42]. A similar result was also observed following freezing-induced dehydration of *A. thaliana* [25].

The exchange of head groups can lead to substantial changes in the composition of different lipid classes and concomitant changes in the physical and biochemical properties of the membrane. PA levels in ECs increased by 66% under desiccation conditions (CL treatments), and by 120% under freezing conditions (CLPFT treatment); whilst decreasing to the level of the control during the recovery process (CLPFTUR) (Figure 7 and Table 1). The accumulation of PA is a common response to the imposition of dehydration and low temperature stresses. During freezing and thawing, PA forms by PLD-catalyzed hydrolysis, and PA increases correspond to molecular decreases in PC and MGDG [25]. Freezing or desiccation stresses induce significant increases in PA levels, and suppression of PA accumulation by the genetic knockdown of phospholipase Dα1 confers relative increases in plant freezing and seed desiccation tolerances [27,34]. Similar changes in the pattern of PA accompanied desiccation in the resurrection plant *Craterostigma plantagineum* [28]. Overall, these results indicated a relationship between PA and desiccation and freezing-induced membrane lesions, suggesting that head group exchange had a role during successful cryopreservation and may have affected the physical and biochemical properties of membranes. However, there are more lipid molecules involved in head group exchange than the 11 classes we analyzed. For example, the phosphate head group of PA could be exchanged with that of DAG or diacylglycerol pyrophosphate (DGPP) [43], and these two types of lipid were not measured in this study.

Once the PA that accumulated during the loading treatment transduced the desiccation stress signal to the cell membrane, total phospholipids (especially PC and PE) significantly increased during the following cryoprotection treatment with PVS2 treatment (CLP) (Figure 7 and Table 1). PVS2 includes four CPAs (DMSO, glycerol, ethylene glycol, and sucrose) which are known to inhibit membrane leakiness during freezing [44]. Our results indicated there might be a lipid–CPA interaction during PVS2 cryopreservation. Studies have often shown that CPAs can interact with the polar head groups of phospholipids, thus forming a layer around the surface of the membrane [45,46,47,48]. Although the cryoprotection mechanism of anti-freezing protein 752 (AFP752) and DMSO revealed different ways to improve the viability of frozen/thawed human skin fibroblasts cells [49], the mechanisms by which different CPAs confer resilience in plant cells need further study.

In conclusion, successful cryopreservation was found to involve increases of total lipids and attenuated lipid degradation in both PVS (Figure 7) and when using controlled rate cooling [3]. But how this outcome was achieved varied with lipid composition and cryopreservation protocols. We proposed two working models to explain how membrane lipid metabolism responds to cryopreservation. When PC is the main lipid component, as in our present study, PA increased with the loading treatment and this might act as a signaling molecule to trigger membrane lipid remodeling in response to this mild osmotic stress. In this case, DAG and PA might be intermediates of lipid metabolism. During the severe chemical dehydration and cryoprotection phase with highly concentrated PVS2, protection was provided through lipid–CPA interaction. These interactions coincided with increases total phospholipids, especially with PC and PE. In contrast, when MGDG was the main lipid component, as in the algal species *C. reinhardtii*, PA did not change during the cryopreservation process and DMSO failed to protect membranes. Instead, MeOH could significantly attenuate lipid degradation, with the TAG pool potentially serving as intermediates rather than DAG [3]. These findings provided deep understanding of the intimate relationship between membrane composition and remodeling and the optimization of cryopreservation procedures with different cell types. Our findings may lead to new technology development in the future for the most recalcitrant of cell systems.

## 4. Materials and Methods

### 4.1. Plant Material and Induction and Maintenance of ECs

*M. officinalis* mature seeds were collected in Hunan, China. Embryo cells (ECs) were induced from mature zygotic embryos. Mature seeds of *M. officinalis* were washed with detergent, surface-sterilized in 0.1% sodium dichloroisocyanurate (NaDCC) solution for 2 h, transferred into 75% ethanol for 2 min in a laminar-flow cabinet, and transferred into 5% NaDCC for 20 min. NaDCC was then drained, and seeds were rinsed three times with sterile deionized water. Zygotic embryos were separated from seeds and placed in Petri dishes containing solid induction medium, (woody plant medium (WPM); 2, 4-dichlorophenoxyacetic acid (2,4-D, 2 mg/L); 6-benzylamino purine (BA, 0.25 mg/L); polyvinylpyrrolidone (PVP, 1 g/L); sucrose (4%); and Phytagel (0.3%), pH 5.8). ECs were maintained in the dark at 25 ± 1 °C.

Calluses were induced after subculture of zygotic embryos three times (at 4-week intervals). A stereomicroscope was used to separate embryogenic from non-embryogenic cells. ECs were transferred into purification medium (WPM; 2,4-D (1 mg/L); PVP (1 g/L); activate charcoal (AC, 1 g/L); sucrose (3%); and phytagel (0.3%), pH 5.8) ECs were obtained after a purification process of two subcultures (4 weeks each). The ECs were then transferred into maintenance medium (WPM; 2,4-D (1 mg/L); PVP (1 g/L); AC (2 g/L); casein hydrolysate (CH, 1 g/L); sucrose (30 g/L), pH 5.8) for cryopreservation.

### 4.2. Cryopreservation Protocols for ECs of M. officinalis

The cryopreservation protocol was based on the work of Liang (unpublished). The ECs of *M. officinalis* were transferred into 2-mL cryovials (0.25 mL volume EC per cryovial) after 2 weeks on maintenance medium. Then 2 mL loading solution (WPM, 2 M glycerol, and 0.4 M sucrose) was added to each, and vials were incubated at 25 °C for 20 min. Loading solution was replaced by 2 mL PVS2 (at 0 ± 1 °C and incubated for 15 min and then vials were plunged directly into liquid nitrogen overnight. PVS2 contained 30% glycerol (*w/v*), 15% ethylene glycol (*w/v*), and 15% dimethylsulfoxide (DMSO; *w/v*). After freezing overnight, cryovials were thawed in a water bath at 40 ± 1 °C for 2 min. PVS2 was replaced by 2 mL unloading solution (WPM and 1.2 M sucrose) and vials were incubated at 25 °C for 20 min. Subsequently, ECs were rinsed 2 times with WPM solution, transferred to maintenance medium in clumps, and placed in the dark for 2 weeks. Three samples (replicates) were analyzed to determine viability after each cumulative step of the cryopreservation protocol (Figure 1): (1) C (control, normal growth); (2) CL, (control + loading); (3) CLP, (control + loading + PVS2); (4) CLPFT, (control + loading + PVS2 + freezing in LN + Thawing); (5) CLPFTU, (control + loading + PVS2 + freezing in LN + thawing + unloading); and (6) CLPFTUR, (control + loading + PVS2 + freezing in LN + thawing + unloading + recovering). Samples (five replicates) were also taken after each stage for lipid analysis. The other six treatments each lacked a certain step of the cryopreservation protocol and were sampled after 2 weeks of recovery; thus, treatment (7) CLPFTUR/-L lacked the 20-min loading step; (8) CLPFTUR/-L+P lacked the 20-min loading step, but had a 35-min PVS2 step; (9) CLPFTUR/-P lacked the 15-min PVS2 step; (10) CLPFTUR/-P+L lacked the 15-min PVS2 step, but had a 35-min loading step; (11) CLPFTUR/-FT lacked the freezing/thawing step; and (12) CLPFTUR/-U lacked the unloading step.

### 4.3. Measurement of ECs Viability

Fluorescein diacetate (FDA, Sigma–Aldrich, St. Louis, MO, USA) staining was used to measure the viability of ECs after various cryopreservation treatments. Embryogenic cells (0.25 mL) were stained for 5 min with a solution of 1 mL WPM solution + 2 uL FDA (5 mg/mL) in DMSO. Thereafter, the cells were rinsed three times with WPM solution and observed under a confocal laser scanning microscope (FV-1000; Olympus, Japan). Green fluorescence under excited light (488 nm) and emitted light (520 nm) signified living tissue, while dead tissue remained unstained. ECs’ viability during cryopreservation was assessed after each process (Figure 1, Treatments 1–6), and after 2 weeks recovery (Treatments 7–12). Three samples from each treatment were observed under the confocal microscope, and fluorescence intensities were recorded.

### 4.4. Lipid Extraction and ESI-MS/MS Analysis

The processes of lipid extraction, ESI-MS/MS analysis, and quantification were performed as described previously, with minor modifications [34]. In brief, ECs from one Petri dish were harvested by carefully separating out from the culture medium at sampling time. To inhibit lipolytic activity, ECs were transferred immediately into 3 mL of isopropanol with 0.01% butylated hydroxytoluene at 75 °C. The samples were extracted by 2 days of agitation in 4 mL chloroform/methanol (2:1, *v/v*, with 0.01% butylated hydroxytoluene). All remaining plant tissue was heated overnight at 105 °C for 18 h and weighed to give the dry weight of each sample. Data processing was performed as previously described [27]. The lipids in each class were quantified by comparison with two internal standards of the class. Five replicates of each cryopreservation treatment were analyzed. Lipid samples were analyzed on a triple quadrupole tandem mass spectrometer (API 4000, Applied Biosystems, Foster City, CA, USA) equipped for ESI. The lipids in each class were quantified in comparison to the two internal standards of that class, using a correction curve determined between standards [34]. The internal standards include the following: di 14:0-PG (0.25 nmol/mL), di 20:0-PG (0.25 nmol/mL), di 12:0-PE (0.25 nmol/mL), di 23:0-PE (0.25 nmol/mL), di 12:0-PC (0.50 nmol/mL), di 24:1-PC (0.50 nmol/mL), di 14:0-PA (0.25 nmol/mL), di 20:0-PA (0.25 nmol/mL), di 14:0-PS (0.17 nmol/mL), di 20:0-PS (0.17 nmol/mL), 16:0–18:0-PI (0.20 nmol/mL), di 18:0 PI (0.14 nmol/mL), 34:0-MGDG (1.67 nmol/mL), 36:0 MGDG (0.33 nmol/mL), 34:0 DGDG (0.41 nmol/mL), and 36:0 DGDG (0.59 nmol/mL). One-way analysis of variance (ANOVA) in combination with Fisher’s least significant difference (LSD) method was used. Heat map analysis was performed using Genespring version 7.2 (Silicon Genetics).

### 4.5. Statistics Procedures

The data were subjected to one-way analysis of variance (ANOVA) with SPSS 16.0. Statistical significance was tested by Fisher’s least significant difference (LSD) method. Hierarchical clustering analysis was performed using Cluster 3.0 and Java Tree-View. Principal component analysis was conducted with SPSS 16.0. DBI and ACL were calculated as described previously [37]: DBI = (Σ[N × mol %lipid)]/100, where N is the number of double bonds in each lipid molecule; ACL = (∑[n × mol% lipid])/100, where n is the number of acyl carbons in each lipid molecule.

## Figures and Tables

**Figure 1 ijms-22-02174-f001:**
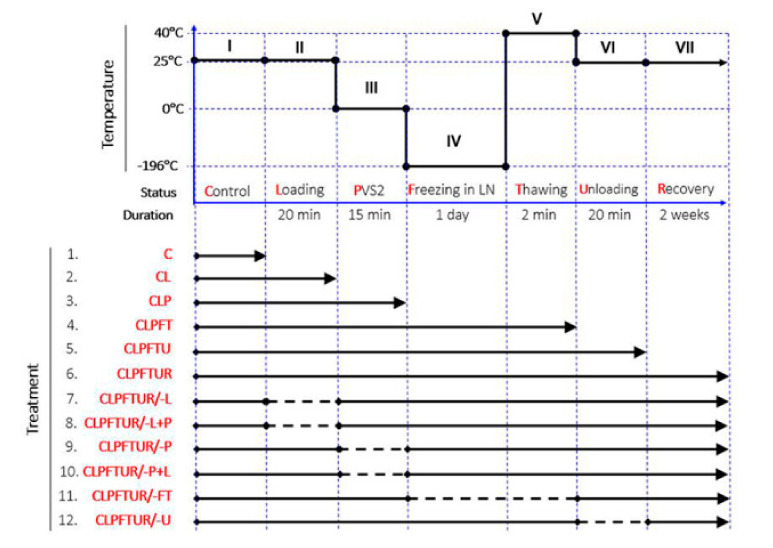
Steps in the cryopreservation process for embryogenic cell aggregates (ECs) of *Magnolia officinalis*. (I) Control (normal growth) (C); (II) loading treatment (osmoprotection) for 20 min at room temperature (L); (III) plant vitrification solution 2 (PVS2) treatment (dehydration with vitrification solution) for 15 min at 0 °C (P); (IV) freezing in liquid nitrogen (rapid cooling) for 1 day (F); (V); thawing (rapid warming) at 40 °C for 2 min (T); (VI) unloading treatment (rehydration and dilution of the vitrification solution) for 20 min at room temperature (U); and (VII) recovery growth for 2 weeks in normal growth conditions (R). To separate out the effects of individual steps in the overall cryopreservation process, single steps were excluded from the protocol (Treatments 1–6). Samples were also taken from six other condition combinations (Treatments 7–12) but with one step missing: (−L) no loading treatment; (−L+P) no loading treatment, but extended PVS2 treatment time (35 min); (−P) no PVS2 treatment; (−P+L) no PVS2 treatment, but extended loading treatment (35 min); (−FT) no freezing and thawing; and (−U) no unloading treatment.

**Figure 2 ijms-22-02174-f002:**
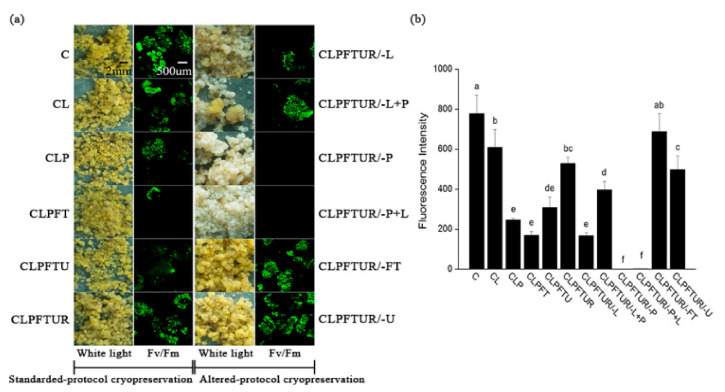
Viability counts of *Magnolia officinalis* embryogenic cells (ECs) during standard- and altered-protocol cryopreservation. (**a**) Viable ECs show green fluorescence after fluorescein diacetate (FDA) staining, while dead tissues remain unstained. (**b**) Photographs are representative of three independent replicates in each case. Fluorescence intensities were recorded, and values are mean ± SD (*n* = 3). Explanations of the notations used for each step are provided in full in the legend to Figure 1. Columns with different letters are significantly different (*p* < 0.05).

**Figure 3 ijms-22-02174-f003:**
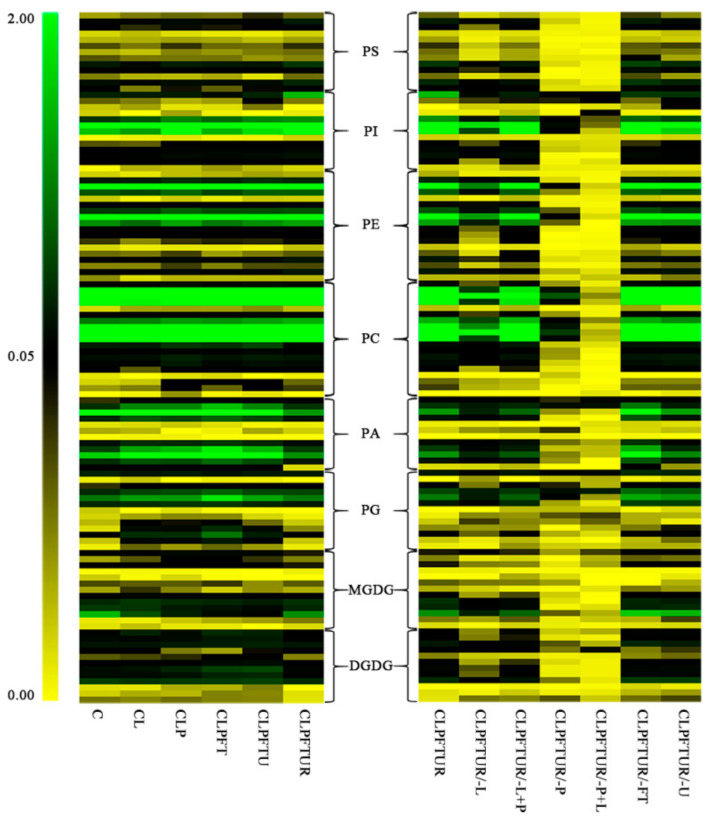
Effect of cryopreservation processes on the phospholipid profile of *Magnolia officinalis* embryogenic cell (ECs). Left panel: Standard-protocol cryopreservation. Right panel: Altered-protocol cryopreservation. Each colored bar within a column represents a lipid species in the indicated treatments (see Figure 1 legend for notation). The color of each bar represents the level of the corresponding lipid species (nmol/mg dry weight). A total of 113 lipid species in the indicated lipid classes are organized using class (as indicated), total acyl carbons (in ascending order within a class), and total double bonds (in ascending order within a class and total acyl carbons). The dry weight is the dry weight of tissue after lipid extraction.

**Figure 4 ijms-22-02174-f004:**
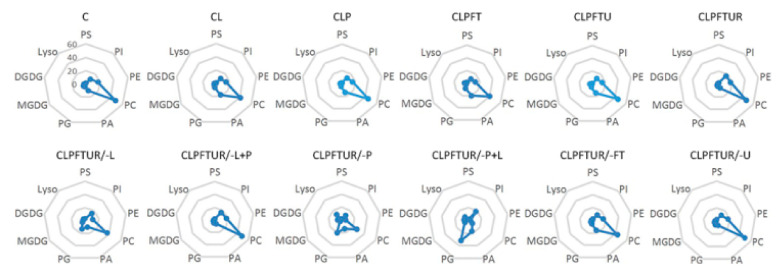
Molecular percentages of lipid classes in *Magnolia officinalis* ECs after cryopreservation treatments. Twelve different treatments (see Figure 1 legend for notation) were examined and compared. Values are means of *n* = 5.

**Figure 5 ijms-22-02174-f005:**
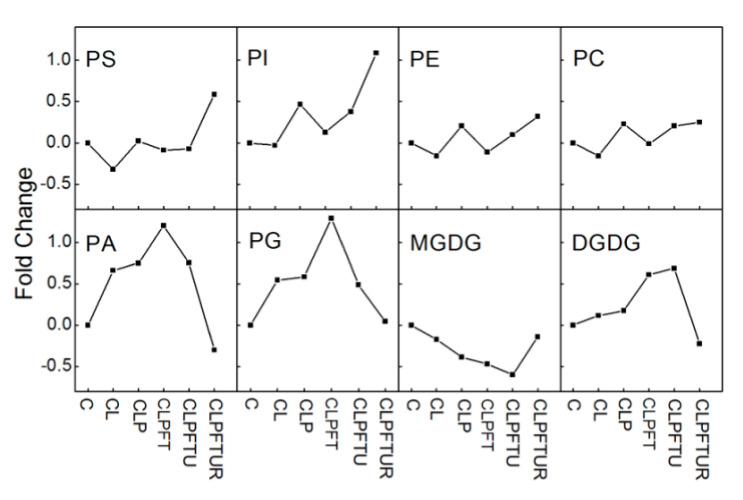
Changes in levels of eight phospholipids in *Magnolia officinalis* ECs during cryopreservation. The fold changes were calculated using the average amount of total lipids of each class (*n* = 5) with the formula (A2 − A1)/A1, where A2 is the total lipid class amount for the indicated process (notation in Figure 1 legend) and A1 is the total lipid amount in the control.

**Figure 6 ijms-22-02174-f006:**
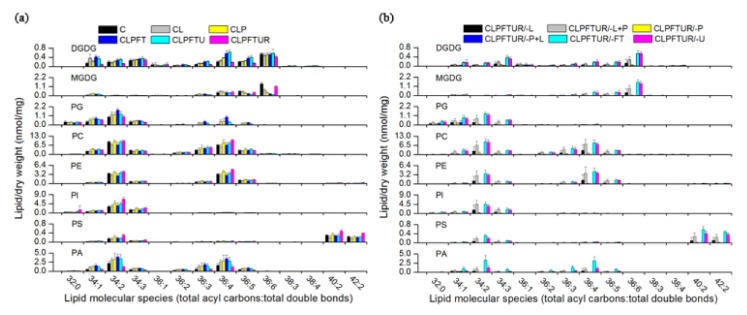
Changes in levels (nmol/mg) of the molecular species of galactolipids and phospholipids during various *Magnolia officinalis* EC cryopreservation protocols. (**a**) Standard-protocol cryopreservation steps. (**b**) Altered-protocol cryopreservation. See Figure 1 legend for treatment notation. Values are mean ± SD (*n* = 5).

**Figure 7 ijms-22-02174-f007:**
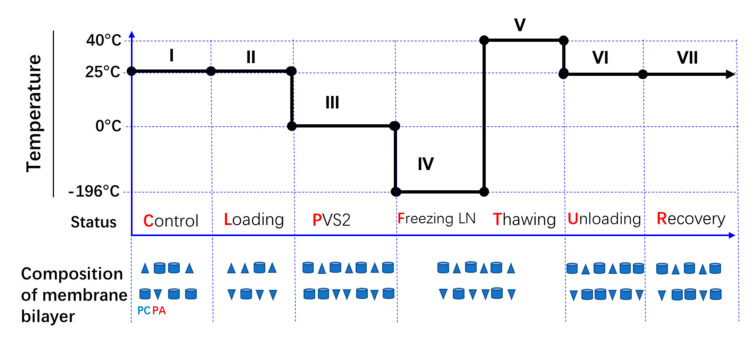
Schematic representation of membrane composition changes in *Magnolia officinalis* ECs during cryopreservation. The columns represent phosphatidylcholine (PC) and phosphatidylethanolamine (PE), while the triangles represent phosphatidic acid (PA). The number of columns and triangles represents the total amount of lipids.

**Table 1 ijms-22-02174-t001:** Total amount of lipid (nmol/mg dry weight) in each head-group class in *Magnolia officinalis* ECs after various cryogenic treatments. Values in the same column marked with different letters are significantly different (*p* < 0.05). Values are means ± SD (*n* = 5).

	**Total PS**	**Total PI**	**Total PE**	**Total PC**	**Total PA**	**Total PG**	**Total MGDG**	**Total DGDG**	**Total Lipid**
C	1.08 ± 0.046 ^de^	5.89 ± 0.69 ^bc^	10.1 ± 0.41 ^abc^	27.98 ± 1.2 ^abc^	6.04 ± 1.68 ^b^	2.28 ± 0.20 ^c^	2.64 ± 0.5 ^a^	1.76 ± 0.32 ^bc^	58.39 ± 2.14 ^cd^
CL	0.73 ± 0.27 ^ef^	5.72 ± 1.58 ^cd^	8.51 ± 1.67 ^c^	23.61 ± 5.44 ^c^	10.04 ± 1.91 ^a^	3.52 ± 0.94 ^b^	2.19 ± 0.32 ^ac^	1.97 ± 0.34 ^b^	57.29 ± 10.75 ^de^
CLP	1.1 ± 0.33 ^de^	8.65 ± 2.03 ^b^	12.2 ± 0.93 ^ab^	34,42 ± 7.1 ^a^	10.56 ± 5.37 ^a^	3.61 ± 0.86 ^b^	1.62 ± 0.54 ^cd^	2.07 ± 0.14 ^b^	75.60 ± 8.89 ^a^
CLPFT	0.98 ± 0.13 ^de^	6.66 ± 2.42 ^bc^	8.97 ± 2 ^bc^	27.72 ± 6.93 ^abc^	13.30 ± 3.05 ^a^	5.22 ± 0.57 ^a^	1.40 ± 0.26 ^d^	2.84 ± 0.48 ^a^	68.33 ± 8.52 ^abc^
CLPFTU	1.00 ± 0.24 ^de^	8.11 ± 1.61 ^bc^	11.11 ± 1.3 ^abc^	33.72 ± 4.82 ^ab^	10.60 ± 3.68 ^a^	3.40 ± 0.54 ^b^	1.06 ± 0.11 ^de^	2.97 ± 0.62 ^a^	73.21 ± 9.47 ^ab^
CLPFTUR	1.71 ± 0.15 ^ab^	12.3 ± 2.9 ^a^	13.34 ± 0.33 ^a^	34.99 ± 1.71 ^a^	4.22 ± 1.22 ^bc^	2.39 ± 0.28 ^c^	2.27 ± 0.23 ^ac^	1.37 ± 0.18 ^c^	73.35 ± 4.61 ^ab^
CLPFTUR/−L	0.46 ± 0.58 ^f^	3.03 ± 3.78 ^de^	3.44 ± 5.02 ^d^	9.14 ± 11.87 ^d^	1.59 ± 2.12 ^cd^	1.03 ± 0.74 ^d^	0.69 ± 0.77 ^e^	0.45 ± 0.48 ^de^	20.10 ± 25.34 ^f^
CLPFTUR/−L+P	1.14 ± 0.52 ^cd^	8.27 ± 3.13 ^bc^	10.12 ± 5.6 ^abc^	25.70 ± 10.9 ^bc^	3.11 ± 1.7 b^cd^	2.20 ± 0.89 ^c^	1.39 ± 0.61 ^d^	0.82 ± 0.43 ^d^	43.49 ± 8.52 ^e^
CLPFTUR/−P	0.06 ± 0.02 ^g^	0.49 ± 0.43 ^e^	0.38 ± 0.4 ^de^	2.39 ± 3.07 ^de^	0.41 ± 0.21 ^d^	0.58 ± 0.12 ^d^	0.11 ± 0.08 ^e^	0.16 ± 0.03 ^e^	5.25 ± 4.86 ^g^
CLPFTUR/−P+L	0.05 ± 0.03 ^g^	0.37 ± 0.22 ^e^	0.07 ± 0.02 ^e^	0.13 ± 0.03 ^e^	0.34 ± 0.17 ^d^	0.65 ± 0.32 ^d^	0.11 ± 0.03 ^e^	0.13 ± 0.07 ^e^	1.99 ± 0.55 ^g^
CLPFTUR/−FT	2.08 ± 0.31 ^a^	8.36 ± 2.37 ^bc^	11.99 ± 3.35 ^ab^	32.39 ± 7.64 ^ab^	10.77 ± 3.77 ^a^	3.43 ± 0.24 ^b^	2.64 ± 0.55 ^a^	1.70 ± 0.36 ^bc^	73.94 ± 10.60 ^a^
CLPFTUR/−U	1.52 ± 0.25 ^bc^	6.80 ± 1.63 ^bc^	10.37 ± 1.2 ^abc^	29.80 ± 5.45 ^abc^	3.85 ± 1.51 ^bc^	3.20 ± 0.46 ^b^	2.53 ± 1.02 ^a^	1.67 ± 0.65 ^bc^	60.21 ± 7.74 ^bcd^
	**Total LysoPG**	**Total LysoPC**	**Total LysoPE**						
C	0.08 ± 0.04 ^b^	0.40 ± 0.11 ^cde^	0.13 ± 0.06 ^def^						
CL	0.14 ± 0.05 ^b^	0.58 ± 0.08 ^abcd^	0.29 ± 0.05 ^a^						
CLP	0.23 ± 0.1 ^a^	0.82 ± 0.48 ^a^	0.31 ± 0.12 ^a^						
CLPFT	0.29 ± 0.009 ^a^	0.68 ± 0.15 ^abc^	0.27 ± 0.02 ^ab^						
CLPFTU	0.23 ± 0.06 ^a^	0.74 ± 0.13 ^ab^	0.28 ± 0.07 ^a^						
CLPFTUR	0.12 ± 0.01 ^b^	0.43 ± 0.08 ^bcde^	0.22 ± 0.05 ^abc^						
CLPFTUR/−L	0.10 ± 0.05 ^b^	0.12 ± 0.15 ^ef^	0.06 ± 0.07 ^efg^						
CLPFTUR/−L+P	0.1 ± 0.04 ^b^	0.42 ± 0.33 ^bcde^	0.15 ± 0.13 ^cde^						
CLPFTUR/−P	0.13 ± 0.05 ^b^	0.49 ± 0.6 ^abcd^	0.05 ± 0.03 ^fg^						
CLPFTUR/−P+L	0.08 ± 0.02 ^b^	0.04 ± 0.02 ^f^	0.02 ± 0.02 ^g^						
CLPFTUR/−FT	0.08 ± 0.03 ^b^	0.34 ± 0.13 ^def^	0.18 ± 0.09 ^bcd^						
CLPFTUR/−U	0.08 ± 0.03 ^b^	0.29 ± 0.05 ^def^	0.10 ± 0.02 ^defg^						

## Data Availability

Not applicable.

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
