# Peer review of "Lipid Remodeling Confers Osmotic Stress Tolerance to Embryogenic Cells during Cryopreservation"

_ijms, 2021, doi:10.3390/ijms22042174_

Round 1

Reviewer 1 Report

The paper of Lin, et al. “Lipid Remodeling Confers Osmotic Stress Tolerance to Embryogenic Cell During Cryopreservation” aimed to investigate the influence of plant vitrification solutions on the viability of Magnolia officinalis embryogenic cells after the cryopreservation. In general, the manuscript is well written, understandable, and clear but I would like to make some remarks about the paper presentation. 

  • Some abbreviations are not adequately explained - see line 17 - PA and PG, PC and PE; see line 53 - what are the main difference between PVS and PVS2; 
  • Line 172 - A total of 113 lipid species ... - I can't find the description of 113 lipid species neither in the manuscript nor in the supplementary;
  • Lines 316-326 - can't find the whole procedure of LC-MS/MS lipid analysis, only two notations - ... as described previously ... [35] and ... as previously described [54];
  • Reference Section: I see the careless description of the Section; please see Instruction for Authors and don't forget about uniformity and DOI data.

Finally, the manuscript may be accepted for publication after minor revision.

Author Response

The paper of Lin, et al. “Lipid Remodeling Confers Osmotic Stress Tolerance to Embryogenic Cell During Cryopreservation” aimed to investigate the influence of plant vitrification solutions on the viability of Magnolia officinalis embryogenic cells after the cryopreservation. In general, the manuscript is well written, understandable, and clear but I would like to make some remarks about the paper presentation.

Point 1: Some abbreviations are not adequately explained - see line 17 - PA and PG, PC and PE; see line 53 - what are the main difference between PVS and PVS2;

Response 1: We thank referee for his/her careful reading, the abbreviation part was added in the end of the manuscript. Please see the new text. The main difference between PVS and PVS2 is: Plant Vitrification Solution (PVS) is a general conception refers to all the solutions that could facilitate the formation intracellular aqueous glasses in plants. PVS2 and PVS3 are the most two common solutions used at the moment. PVS2 was first designated by Dr. Sakai in 1990 that contains 30% glycerol (w/v), 15% ethylene glycol (w/v), and 15% dimethylsulfoxide (DMSO; w/v) in basal culture medium (with-out growth regulators) containing 0.4 M sucrose (pH 5.8), and PVS3 consists of 40% glycerol (w/v) and 40% sucrose (w/v) in basal medium (Sakai, et al. 2008).

Point 2: Line 172 - A total of 113 lipid species ... - I can't find the description of 113 lipid species neither in the manuscript nor in the supplementary.

Response 2: The detail data for 113 lipid species used for Fig. 3 and Fig.S1 were provided as supplementary data.

Point 3: Lines 316-326 - can't find the whole procedure of LC-MS/MS lipid analysis, only two notations - ... as described previously ... [35] and ... as previously described [54];

Response 3: The detail description of lipid analysis was provided in the new text. Please see the method and material part 4.4.

Point 4: Reference Section: I see the careless description of the Section; please see Instruction for Authors and don't forget about uniformity and DOI data.

Response 4: We thank referee for his/her careful reading and feel really sorry that we haven’t follow the instruction for authors carefully before. The mistakes have been corrected accordingly; please see the following changes in the reference part of new text.

Reviewer 2 Report

The work contains and describes important material processes in the area
of plant species cryopreservation. Authors took a lipidomics approach using electrospray ionization
mass spectrometry lipid profiling which enabled the quantification of
changes in the molecular species of membrane lipids during cryopreservation.
The authors showed that the successful cryopreservation involved increases of total
lipids and attenuated lipid degradation.
The whole work should be written
in a much clearer way - a number of details are not necessarily stated
in the main text, but in the supplementary mat, appendix. The whole
manuscript will then be
clearer.
The cryoprotection changes of investigated materials
should be presented and explained in correlation with the whole plant
cryoprotection strategy. Also the standard cryoprotective materials effect
(DMSO) should be discussed (I. Kratochvilova et al. Theoretical and
experimental study of the antifreeze protein AFP752,
trehalose and dimethyl sulfoxide cryoprotection mechanism:
correlation with cryopreserved cell viability,
RSC Advances,
7 (1), 352-360).  

Author Response

The work contains and describes important material processes in the area of plant species cryopreservation. Authors took a lipidomics approach using electrospray ionization mass spectrometry lipid profiling which enabled the quantification of changes in the molecular species of membrane lipids during cryopreservation. The authors showed that the successful cryopreservation involved increases of total lipids and attenuated lipid degradation.

Point 1: The whole work should be written in a much clearer way - a number of details are not necessarily stated in the main text, but in the supplementary mat, appendix. The whole manuscript will then be clearer.

Response 1: We thank referee for his/her careful reading and gave such constructive suggestion. We already made some changes to make the paper clearer. For example,Fig. 5 and Table 2 already been moved into supplementary data, and the description of those data were shortened accordingly. The results 2.7 has also been deleted and combined with 2.2. The details of changing please see the new text.

Point 2: The cryoprotection changes of investigated materials should be presented and explained in correlation with the whole plant cryoprotection strategy. Also the standard cryoprotective materials effect (DMSO) should be discussed (I. Kratochvilova et al. Theoretical and experimental study of the antifreeze protein AFP752, trehalose and dimethyl sulfoxide cryoprotection mechanism: correlation with cryopreserved cell viability, RSC Advances, 7 (1), 352-360). 

Response 2: It was a very good point to study the cryoprotection changes with the whole plant cryoprotection strategy. We already mentioned the cryoprotction strategy changes during the development of plant cryopreservation history, e.g. in the second and third paragraph of the introduction section. In the present study, we discovered treatment of PVS2 which consists of four CPAs (DMSO, glycerol, ethylene glycol sucrose), changed membrane composition, but how each CPA, especially DMSO worked on our plant materials physically and/or biochemically is not our focus. We agreed that the standard cyroprotective materials such as DMSO is very important for the cryoprotection. We add related discussion and the reference as the suggested (please see reference 52).

Round 2

Reviewer 2 Report

To be published.